# Reversible bending of U-shaped plant petioles under dehydration

Anne Schliebach[1], Mohammad Nadim Kamar[1], Baptiste Bordet[1], Catherine Quilliet[1], Benjamin Dollet[1], Eric Badel[2], Emmanuel Siéfert[1] and Philippe Marmottant[1] 

[1]Université Grenoble Alpes, CNRS, LIPhy, 38000 Grenoble, France; [2]Université Clermont Auvergne, INRAE, PIAF, 63000 Clermont-Ferrand France

## Original Research Article

biomechanics; nonlinear; slender mechanics; water loss.

**Corresponding author:**
Philippe Marmottant;
Email:
philippe.marmottant@univ-grenoble-alpes.fr

**Associate Editor:**
Dr. Naomi Nakayama

## Abstract

The shape of plants can be sensitive to dehydration. Here, we focus on herbaceous plants whose petiole cross-section is U-shaped and contains a lot of water. Among a large range of plants showing the same behaviour, we examine *Spathiphyllum* that exhibits a pronounced, sudden but reversible drooping under dehydration. We show that it is the consequence of a high-amplitude hinge mechanism located at the base of its long petioles, similar to a carpenter's tape folding under sufficient load. Mechanical testing demonstrated that small-amplitude bending rigidity decreases by only a factor of three during dehydration, due to tissue shrinkage rather than material softening. The petiole is composed of water-rich parenchyma tissue: drooping occurs abruptly at 35%–40% of mass loss, remaining reversible unless dehydration is prolonged. Inspired by these observations, we introduce a biomimetic hinge which offers a programmable bending stiffness and nonlinear behaviour under load, with applications in computing mechanical metamaterials.

## 1. Introduction

The responses of plants to water stress are especially noticeable in herbaceous plants that are non-lignified. Dehydration, or the loss of water mass, induces a drop of the turgor pressure in cells. In non-lignified plants, plant tissues are less stiff, and noticeably deform with a much larger amplitude compared to woody plants. We remind the reader that internal turgor pressure is not only important for the swelling of tissues but is also the essential source of growth (Dumais, 2021). Turgor pressure is an osmotic effect: when the external water potential is high, due to a humid environment, the water flows into the cell through its membrane because the water potential is there lowered by the presence of osmolites. This results in the cell swelling and the building of a turgor pressure with a magnitude of a MPa (Dumais & Forterre, 2012). Dehydration can result in shape changes commonly observed on specific parts of plants such as the palm leaves (Guo et al., 2024) presenting a bilayer effect (Timoshenko, 1925) on its folds, on the stems of resurrection plants (Rafsanjani et al., 2015), on the scales of pine cones (Reyssat & Mahadevan, 2009), on the seeds of Erodium (Aharoni et al., 2012), on the long petioles of of Caladium petioles (Caliaro et al., 2013) or on the Gerbera peduncles (Lehmann et al., 2019).

A common interpretation of the shape change of plants is that a decrease in turgor pressure results in a loss of tissue rigidity, as observed on pieces of potato tubercule (Falk et al., 1958). Mechanical models at the cell scale were developed, elaborating on the fact that a local deformation of a spherical shell under pressure is more difficult when it is pressurised above a threshold, as show when a point force (Vella et al., 2012) or a spherical indenter is pushed in (Couturier et al., 2022). The same reasoning was developed at the tissue scale (Nilsson et al., 1958; Warner et al., 2000), where models assume that rigidity has a part linked to the bending of solid cell walls plus a part due to pressure. The influence of solid properties and internal pressure was comforted on pneumatic cellular bioinspired materials (Tadrist et al., 2022). However, the effect of pressure on the bending rigidity $B$ was found to be counter-intuitively negligible on elongated tubular shapes for small deformations (Haseganu & Steigmann, 1994; Qiu et al., 2022; Siéfert et al., 2019), it is only at large deformations that pressure has an effect by counteracting buckling.

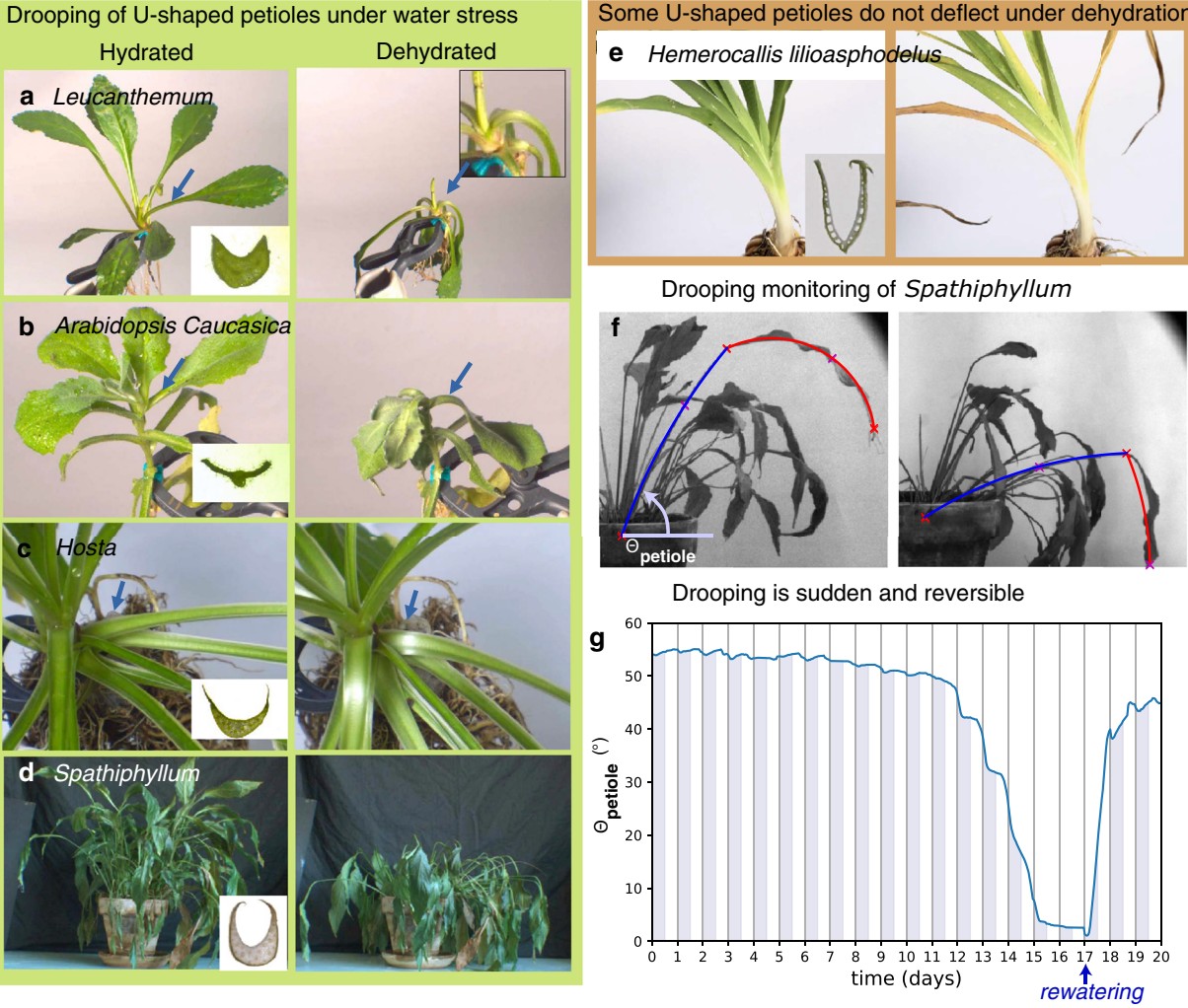

**Figure 1.** Behaviour of the petiole under water stress. (a)–(e) Photographs of five plants with a U-shaped petioles when turgid (left column), and when drying (right column). Plants shown in (a)–(d) feature a strong localised bend (highlighted by an arrow) when dehydrated, whereas plant in (e) does not exhibit any significant shape change. The five plants are: (a) *Leucanthemum* (family of Asteraceae, common name Daisy), (b) *Arabidopsis caucasica* (Brassicaceae), (c) *Hosta* (Asparagaceae), (d) *Spathiphyllum* (Araceae) and (e) *Hemerocallis lilioasphodelus* (Asphodelaceae). Insets show the U-shaped cross-section. (f) Side view of the *Spathiphyllum* plant under dehydration: Natural intelligence (NI) monitoring of the position of the characteristic points of the petiole and leaf (red crosses) and definition of an angle for the petiole on the watered (left image) or dehydrated plant (right image). (g) Evolution of the angle of the petiole with respect to the horizontal, time $t = 0$ at 5 pm, start of the period without watering, rewatering on day 17. Shadowed bars represent nights.

At the plant scale, on elongated parts such as the peduncle of flowers (Lehmann et al., 2019) there can be a dramatic change of the global shape under drought. If this change is due to a decline in the global bending modulus $B$, the softening of the local tissue was found to be weak, the tissue softening being monitored by the effective Young's modulus $E_{\text{eff}} = B/I$, where $I$ is the second moment of area.

There is thus a dichotomy between the change of shape of the plant and the limited softening of its tissue. The solution to this apparent paradox is well described on a recent paper by Chandler et al. (2025), who showed on cellular sheet that pressure controls the geometry of the cross-section and therefore the inflation of the cross section area. The apparent bending rigidity of the plant indeed undergoes a 'geometrical' stiffening: at large pressures, the cross-section size increases, resulting in a global stiffer response to external loads since it increases the second moment of area $I$.

Here, we focus on the behaviour of the petiole, which links the leaf to the stem. In addition to its functional role – holding the leaf, and conveying the sap – observations reveal that in many species the petiole wilting also protects the leaf against damages by direct sun and may protect against water stress (Chiariello et al., 1987; Gonzalez-Rodriguez et al., 2016; Zhang et al., 2010).

Petioles present a large diversity of shapes (Filartiga et al., 2022): among these, petioles whose cross-section assume a U-shape, also called sulcate or canaliculated (Faisal et al., 2010) are common among several families of plants (including the Arabidopsis genus) (see Figure 1).

The U-shape reinforces the bending stiffness of the petiole, resulting in a more rigid beam for the same amount of material compared to a circular cross-section shape (Ennos et al., 2000). U-shaped structures present a high stiffness to weight ratio, and are ubiquitous in plants, as well as in civil engineering and applied mechanics. The mechanics of large amplitude deformation of U-shaped structures was extensively studied (Barois et al., 2014, 2021; Kumar et al., 2023; Seffen & Pellegrino, 1999; Taffetani et al., 2019; Wei et al., 2023; Wuest, 1954).

To induce dehydration, a drastic procedure is to rinse out the soil and hold plants with elastic clamps and let it dry in air, with roots exposed to air. We observe that in some cases gravity is enough to bend the U-shape petiole when turgor is lost (Figure 1(a)–1(c)). This localised bending is common among species with U-shaped petioles, from very different plant families, dicotyledon (such as *Leucanthemum* and *Arabidopsis caucasica*) or monocotyledon (such as *Hosta*). However, it is not systematic since petioles without water-rich tissues do not bend at all under gravity, leaves remaining at the same position in our drying test (such as *Hemerocallis, Dracaena* or *Vriesa* featuring thin leaves).

The problematics that emerges is: how does wilting occur with a U-shape petiole? Is it a change of the spontaneous curvature with an active bilayer effect (Armon et al., 2011; Reyssat & Mahadevan, 2009), or a passive effect due to the softening of the tissue as initially hypothesised (Nilsson et al., 1958; Warner et al., 2000) or due to the evolution of the cross-section geometry as modelled by Chandler et al. (2025)?

Here, we explore this problematics by looking at the spectacular change of shape of *Spathiphyllum*, a common interior plant from the Araceae family that dramatically changes its appearance under dehydration; a state that is reversible (Figure 1(g)). This non-lignified plant has no stem and consists of a group of leaves, each blade being attached directly to the root by a long petiole. The plant features a U-shaped cross-section only at the base of the petioles.

Our approach is the following: First, we describe the global plant shape and this natural intriguing process. Second, we present the anatomy and the geometry of petiole sections. Third, we carried out bending and mass measurements to characterise the softening of the petiole under drought. Fourth, we focus on the plant part presenting the highest bending: the base, acting as a hinge. We model the non-linear mechanics of the base, inspired by the carpenter's tape. Lastly, possible engineering applications are suggested with a biomimetic model, actuated by pressure.

## 2. Monitoring the plant shape under dehydration

A first observation is that under sustained drought, all petioles became more horizontal, with leaves drooping. However, all lift up in less than one day after rewatering (Figure 1(g) and Movies M0, M1 and M2 in the Supplementary Material. Movie M0 in the Supplementary Material is all-public movie summarising the findings, and Movies M1 and M2 in the Supplementary Material show the shape evolution). We recorded and analysed a sequence of photographs of a petiole and leaf taken during 20 days. The images were too complex to be analysed with artificial intelligence techniques, so we used standard 'natural intelligence (NI)' to monitor the position of the characteristic points of the petiole and leaf, using a human brain to click on points on a sequence of images. The points that we tracked were: base and middle of the petiole, base, middle and extremity of the leaf blade. The result was a sharp decrease in the angular height when unwatered for a long duration (here 17 days; this duration fell to two days when the roots were removed from the soil). This process was reversible after rewatering, if dehydration was not too prolonged. In the process, the petiole remained quite straight, the global curvature in between the base and the leaf attachment increased by only 50% (Figure S1 in the Supplementary Material). As a side note, the leaves rose up every night but eventually fell vertically in the dehydrated state (Figure S1 in the Supplementary Material).

## 3. Anatomy and rigidity of the petiole

A closer inspection was performed by looking at sections of the different parts of the petioles (Figure 2(a)). At the end of the petiole, there is a junction, the pulvinus, slightly swollen and lighter in colour, which is known to actively orient the leaf. Sections revealed that the pulvinus is roughly cylindrical, as well as the top of the petiole (not shown). On the contrary, the base of the petiole presents a U-shape, with an important cavity on the adaxial side, closer to the central vertical axis. The cavity is filled by a newer growing leaf, and leaves are nested within each other. There is a transition from the base to the top of the petiole, the U-shape evolving in a rounder shape with thin side blades remaining, see middle cut on Figure 2(a).

Transversal cross sections of the excised leaf, 1-mm thick, were cut with a razor blade, placed on a glass slide and installed under a macroscope (Zeiss Axio Zoom.V16) in an air-conditioned room ($T = 20°C$). The camera was driven by the ZEN software (Zeiss, ZEN 3.9) that allowed to record a sequence of images with a time step of 5 minutes during the natural air drying. For each record, a stack of 20 images was recorded at different altitude positions in order to apply an 'extended depth of focus' post-treatment that corrects the non-perfect flatness of the sample. Magnification was ×13.2 for a spatial resolution of 0.24 pixels/$\mu$m.

All the sections reveal that most of the interior tissue is a water-filled parenchyma tissue with large cells. There are also vascular bundles, lignified, see regularly spaced spots on the figure, that can be coloured using a dye (Figure S2 in the Supplementary Material). The interior is protected by a green and dense cuticle.

### 3.1. Mass and width decline

In order to estimate the local evolution of the properties of the plant, we performed a range of measurements on: (i) the whole plant with roots (out of the soil), (ii) a petiole cut at the base and the top extremity, and (iii) small pieces of the petiole at different locations (Figure 2(b)). The dehydration was performed in ambient air, eventually killing and drying completely the plant.

The mass loss can reach more than 90% of the initial mass for the petiole (Figure 2(c) and more than 95% for cut pieces (Figure S3 in the Supplementary Material). Indeed, most of the aerial part of the plant is constituted with water, the interior tissue acting like a reservoir. The mass decay over time is well described by a model with homogeneous diameter that accounts for the fact that exposed surface area of the cut section subject to evaporation is decreasing over time (see Appendix A). The model is fitted adjusting only $k$, the rate constant giving the flux of evaporation through the cut surface. This model better fits data than a model with a cross-section constant over time.

Width measurements were performed using a caliper, from the side (operating from the top would provide less reproducible results due to the thin blades). They featured, like mass measurements, a strong decay of the diameter until complete drying, up to 60% (Figure 2(d). There was no significant change in axial length. This results from the natural structural anisotropy of the vascular fibres made of long cells making them less extensible in the longitudinal direction, while the tissue can easily contract transversely in between fibres.

### 3.2. Bending stiffness evolution

We also performed 3-point bending measurements on the same samples as for mass and width measurements. The outer points

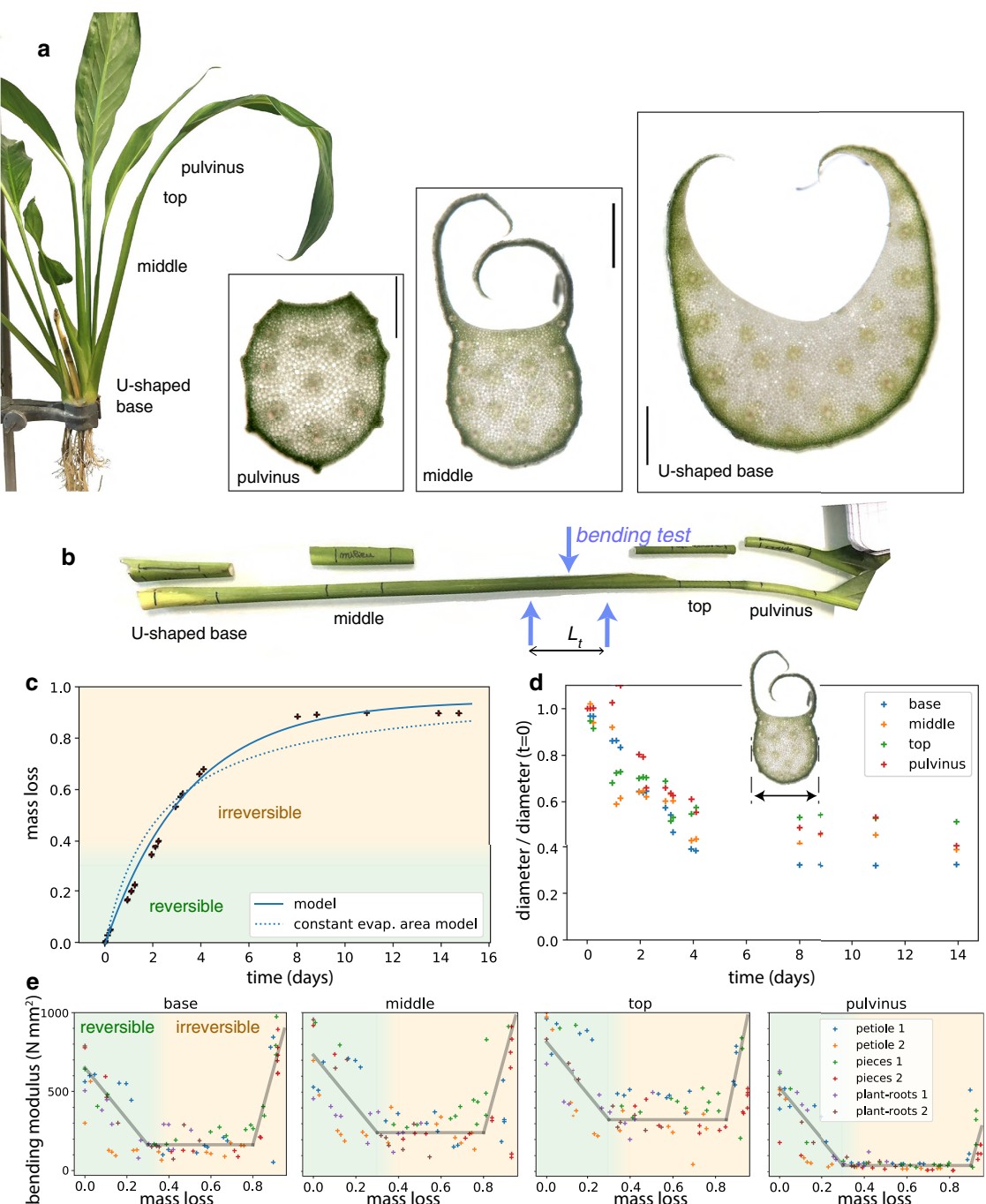

**Figure 2.** Geometrical and mechanical characterisation over dehydration. (a) Anatomy of the petiole. Left: description of the different parts of the petiole. Right: transverse cross-sections, shown at the same scale (bar represents 1 mm). (b) Photograph of cut entire petiole (bottom) and pieces of the petiole (top). Blue arrows depict a 3-point bending test with a support span of $L_t = 25$ mm. (c) Relative mass loss versus time. The relative mass loss is defined as $(m_0 - m)/m_0$, with $m_0$ the initial mass and $m$ the actual mass. Entire petiole without leaf $L = 33.5$ cm. The continuous line represents a fit with a decaying cross-sectional area and evaporation flux $k = 1.0 \times 10^{-6}$ kg/s/m². The dotted line represents a model assuming evaporation through a constant cross-sectional area. The green zone shows the range 0 to 0.3–0.4 where dehydration is reversible, while above irreversible tissue death starts to occur. (d) All the pieces feature a strong diameter loss, while the length remains constant. (e) The bending modulus measured with a bending test follows three phases: a decay, then a plateau for a critical mass loss, and eventually an increase near complete drying. Lines are guides to the eye. Measurements on entire petiole, pieces and whole plant with roots.

were separated by $L_t = 25$ mm and a force $F$ was applied on the centre point. After positioning the sample in a custom 3-point bending setup, we incremented progressively the central position $\delta$ from 0 to 1 mm. The position is changed via a translation stage (Thorlabs Z625B). On this stage is attached a central rod via a 1-N force sensor with a Wheatstone Bridge (Phidget 3139_0). This rod is a 2-mm-thick aluminium plate with a rounded V-shape on the bottom to avoid cutting the stem while applying the force on a small area. The Wheatstone Bridge is sampled with a DAQ (Phidget DAQ1500) at 10 Hz. The synchronisation between the actuator and the sampling is done by a multi-thread homemade Python script. We performed a linear fit to obtain the stiffness $k$, from the relation $F = k\delta$.

**Table 1.** Decay of bending modulus until a plateau, values fitted

|  | Water loss | $B_{plateau}$ | $B_{watered}/B_{plateau}$ |
|---|---|---|---|
|  | at plateau | (N/mm) |  |
| U base | 0.33 | 147 | 3.6 |
| Middle | 0.36 | 231 | 3 |
| Top | 0.44 | 354 | 3 |
| Pulvinus | 0.38 | 39 | 12 |

*Note*: Average values presented, with a standard deviation less than 30% for $B_{plateau}$, and less than 40% for $B_{watered}/B_{plateau}$.

The force was applied from the side of the petiole and not on the adaxial or abaxial side, resulting in side bending in order to avoid contact with the thin blades and get more reproducible measurements. The stiffness $k$ relates the force $F = k\delta$ to small central displacements $\delta$. According to the linear theory of elongated beams, the force is $F = 48B\delta/L_t^3$ (Landau & Lifschitz, 1967), with a bending modulus $B$ that is deduced from the stiffness.

The bending modulus of the petiole follows three phases when mass loss progresses: (i) a decrease, (ii) a long plateau, eventually followed by (iii) a fast increase, a rigidification when the drying is close to final (see Figure 2(e)). The typical water loss required to reach the plateau is given in Table 1 and was around 0.35 (i.e., 35% of the initial mass). The decrease in stiffness was very high in the pulvinus (which explains why leaves droop), but was limited to a factor around 3 in the rest of the petiole.

In the literature, an effective Young's modulus is computed from the bending modulus, using the fact that $B = EI$ for a beam made of a homogeneous material, with $E$ the Young's modulus and the second moment of area $I = \pi(d/2)^4/4$ for round beam of diameter $d$ (Landau & Lifschitz, 1967). Here, although the petiole cannot be considered as a beam made of homogeneous material since it contains different types of tissues, we can define an effective Young's modulus $E_{eff} = B/I$, taking the measured width as the diameter $d$. The measured effective Young's modulus following this method did not seem to decrease on the physiologically reversible range 0–0.4 in water loss, the phase (i) before the plateau (Figure S4 in the Supplementary Material). This echoes the work by Niklas (1991), who included *Spathiphyllum* plants on vibration tests and found a small variation of the effective Young's modulus over dehydration in the physiological range. Similarly, Lehmann et al. (2019) showed that if the bending modulus $B$ of the peduncles of Gerbera flowers decreased between the turgescent and the wilting state there was no change of the effective Young's modulus $E_{eff} = B/I$. Caliaro et al. (2013) found as well a decrease of the effective Young's modulus $E_{eff}$ of 40% at most for Caladium petioles under drought.

The U base shape is not round, but we found that its bending modulus evolved like the rest petiole for small loads (see again Figure 2(e)). A detailed analysis of the second moment of area from the images of a cross section shows that size (cross-section area) is the main driving for this reduction, and not the evolution of the shape (see Figure S5 in the Supplementary Material).

This experimental part indicates that the strong decrease of diameter is the main explanation for the decrease of the bending modulus. But this decrease alone cannot explain the dramatic falling behaviour of petioles.

## 4. High amplitude bending of the base: A hinge

### 4.1. Angular change

Although the overall curvature of the petiole did not increase significantly, there was a strong bending of the base. This was more conveniently observed by taking the whole plant out of soil. The plant was held by tweezers, exposing the root directly to air which speeded the drying. The key observation was that the U-shape of the base opened and marked a clear localised fold (Figure 3(a) and Movie M3 in the Supplementary Material). More precisely, the photographs revealed that the U-shaped petiole unwrapped starting from the base and then the opening progressed further up, before the folding occurred. The folding was localised at the base where the torque applied by the weight of the rest of the plant was maximum (Figure 3(b)).

The angle of the petiole with respect to the horizontal ('angular height'), measured on images taken from the side, suddenly dropped at a critical mass loss (around 0.35 of the initial mass), and even became negative (Figure 3(c)). The change in angle (from the initial to the final value) was large: we observed ranges from 90° up to 120° in angle amplitude. It was observed that a mass loss of around 0.4 was not fatal and coud be reversed by watering, the petiole lifting up as before with yellow stains showing the effect of hydric stress on epidermal cells.

### 4.2. Carpenter's tape analogy and Roman toy model

The U-shaped part became thinner over drying as revealed by the observation of slices (Figure 4(a)). In particular we observed that the thickness of the centre of the U shrunk rapidly (Figure S5 in the Supplementary Material).

In order to simplify the mechanics, we model the base as a U-shaped plate of Young's modulus $E$, width $w$, uniform thickness $h$, parameters that can evolve during drying. The important parameter is the initial transverse curvature of the plate $k_0$, that we monitor using the projected thickness $d$, with a ratio $d/h$ of the order 3–4 for turgid plants. From basic geometry, we have the relation $k_0/\rho \simeq 4d/h$ for small curvatures, with $\rho = h/w^2$ a characteristic curvature.

The U-plate of initial transverse curvature $k = k_0$ is rigidified compared to a flat plate. However, for a sufficient torque, the transverse curvature decreases and a mechanical instability occurs, leading to the sudden bending of the U-plate. This is commonly observed with a metallic carpenter's tape (Figure 4(b)), where $d/h$ is initially of order 60.

This non-linear behaviour can be reproduced by the Roman toy model (Ponomarenko, 2012; Roman, 2024) (see Appendix B). This toy model assumes a uniform transverse curvature $k$ along the width, and therefore approximates the traditional analysis by Wuest (1954) where the shape has not a uniform transverse curvature. The U-shape is flexible in opening, and the transverse curvature $k$ can decrease compared to the initial value, softening the response. The torque needed $M$ to produce a curvature $K$ with this non-linear model is

$$M = \frac{Eh^4}{12w}\frac{K}{\rho}\frac{1 + \beta\left(k_0/\rho\right)^2 + 2\frac{\beta}{\alpha}\left(K/\rho\right)^2 + \frac{\beta^2}{\alpha^2}\left(K/\rho\right)^4}{\left[1 + \frac{\beta}{\alpha}\left(K/\rho\right)^2\right]^2}, \quad (1)$$

where $\beta = 1/60$ and $\rho = h/w^2$. The material can be non-isotropic when the parameter $\alpha$ is different from 1, to account for a transverse bending modulus weaker than the longitudinal bending modulus by a factor $\alpha$ ($< 1$), since the petiole is reinforced by longitudinal bundles. The nonlinear response using the exact but more complex

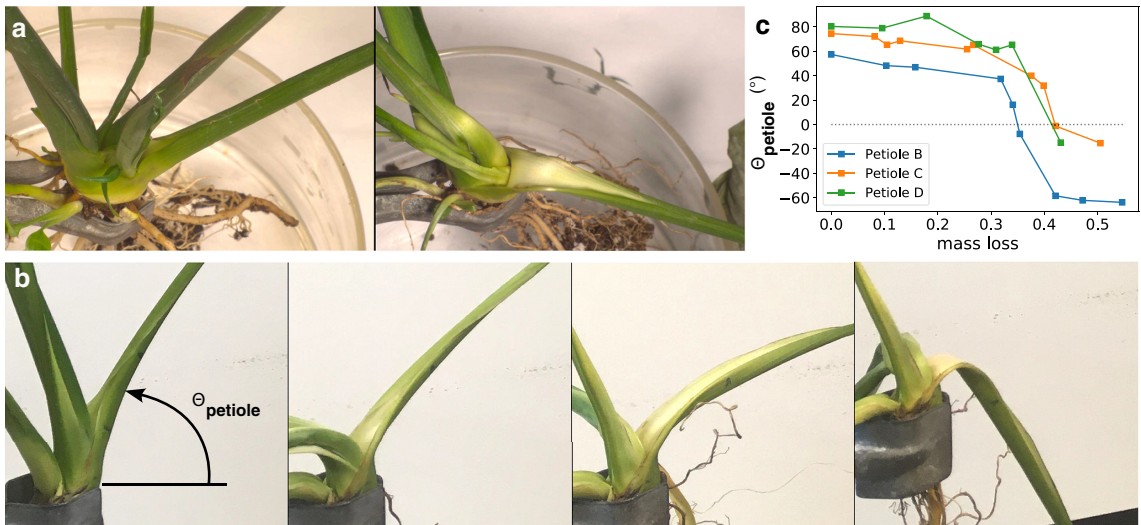

**Figure 3.** Nonlinear hinge at the petiole base. (a) View from above of a plant extracted from soil, in the initial state and then after drying for one day. (b) Another plant (featuring petiole B), seen from the side at times 6.4, 23.2, 25.3 and 47 h, with relative mass loss of 0.16, 0.32, 0.34 and 0.42, showing the opening of the U shape starting from the base, and then the folding. This experiment was reversible after watering. (c) Angle with respect to horizontal versus time, for three petioles.

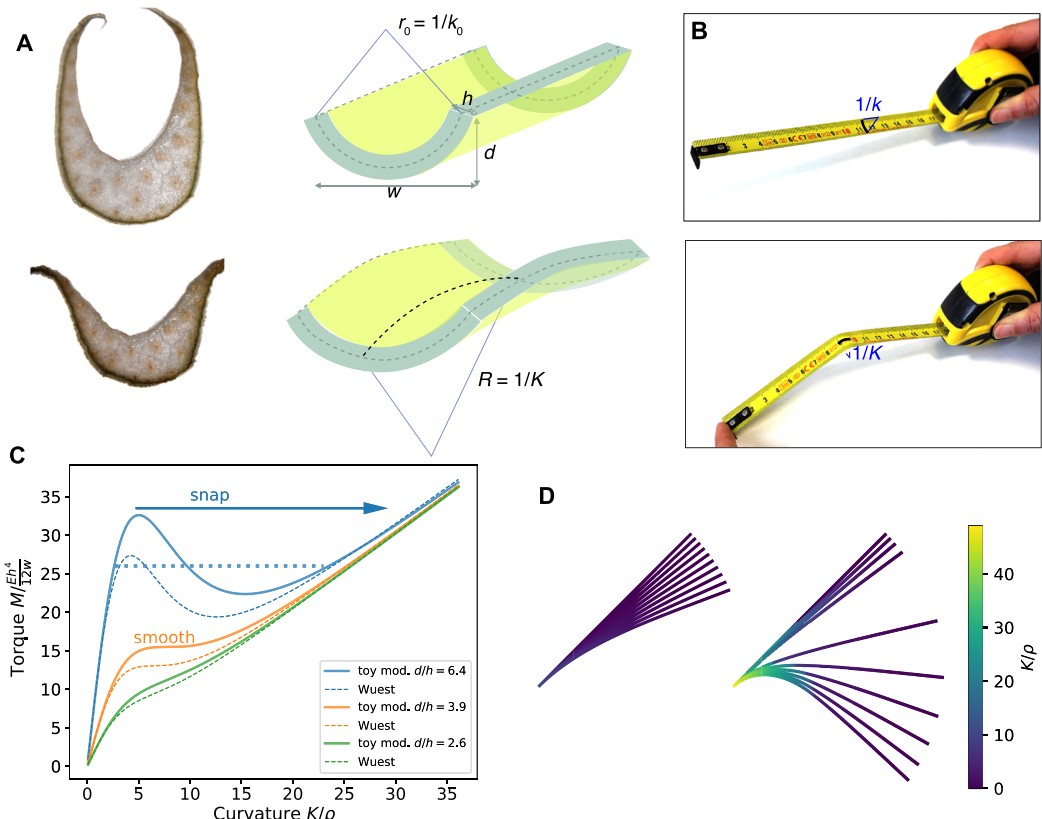

**Figure 4.** Mechanics and analogy with the carpenter's tape. (a) Cross-section of the U-shaped base when hydrated (top) or dehydrated (bottom), showing the decrease in thickness of the $U$. Drawing: mechanical approximation with a constant thickness U-shaped plate. (b) Carpenter's tape with a transverse curvature $k$ (radius of curvature $1/k$, top image). Under torque load, the tape presents a bent region with a longitudinal curvature $K$ (bottom image) while the transverse curvature vanishes there. (c) Non-dimensional torque $12wM/Eh^4$ versus non-dimensional curvature $K/\rho$ for the isotropic toy model (lines, $\alpha = 1$) for several transverse curvature $d/h \simeq k_0/4\rho$, with $\rho = h/w^2$ and for the Wuest model (dotted lines, with $\nu = 0$). (d) Simulation of the shape of a homogeneous beam with constant bending modulus (left) and U-shaped beam with the same bending modulus but with non-linear response described by the toy model (right, $d/h = 3.8$ just below critical point, $L\rho = 0.12$). The linear weight $\mu$ was increased regularly, up to the point that $\mu g L^3 / EI$ reaches 14. Colours indicate local curvature.

predictions from Wuest (1954) (see Appendix C), provides qualitatively similar results.

The curves displaying the non-dimensional torque $M/\frac{Eh^4}{12w}$ as a function of the non-dimensional longitudinal curvature $K/\rho$ feature two regimes depending on the initial transverse curvature parameter $k_0/\rho$: (i) monotonously increasing for transverse curvature smaller than a critical value, such that $d/h \leq (d/h)_c \simeq 3.9$ for the isotropic case $\alpha = 1$ or (ii) presenting a peak for $d/h \geq (d/h)_c$, the peak height increasing with $d/h$ (Figure 4(c)).

In the first regime, monotonous, the bending is stiff at low deformation (high slope on the curve) and then becomes less stiff after an inflection point.

In the second regime, imposing an increasing torque $M$ from zero induces small curvatures until the peak is reached, then leading to a snap. A smaller $\alpha$ compared to 1 does not change the initial slope of the curve but changes the height of the first peak (Figure S6 in the Supplementary Material). After the jump, the tape presents two spatial parts sharing the same torque, one with low curvature $K_1$, and the other with high curvature $K_2$. The Maxwell construction provides the torque value such that the transition from the low to the high value does not generate any work. This condition can be expressed as $\int_{K_1}^{K_2} M \mathrm{d}K = 0$, meaning an equal area above and below the Maxwell plateau (horizontal dotted line of Figure 4(c)). The same phase separation effect, originally developed for phase transitions between different states of matter, appears in the propagating bulges of cylindrical rubber balloons (Kyriakides & Yu-Chung, 1991; Lestringant & Audoly, 2018), and involves a spatial transition (Kumar et al., 2023).

In the plant experiments, the torque $M$ exerted by the weight of leaf did not increase with time. It would even be the opposite since the water mass was decreasing. A decrease of the diameter $d$ (round part of petiole) or thickness $h$ (U-shaped base) at constant petiole length provides a torque scaling as $M \sim d^2 \sim h^2$. However, we expect the non-dimensional torque $M/[Eh^4/12w(1-\nu^2)] \sim h^{-2}$ to increase over time, since $E$ did not seem to vary from our bending measurements.

It is not really clear from section experiments if the U-shape spontaneously opened up in a reference stress-free state during drying, which would mean a decreasing $d/h$ (see photographs of Figure S5 in the Supplementary Material). For the sake of simplicity, we propose the following interpretation: the value of $d/h$ is roughly constant and the evolution of the cross-section geometrically similar. From the model behaviour, we deduce that its value is slightly lower than the critical value 3.9 in experiment, meaning a smooth transition (as observed on time lapse series) and no jump as for a metallic tape. This value seems plausible when looking at the plant cross-section, but exact calculations would be needed to account for the specific geometry and the anisotropy.

As a conclusion, we demonstrate that the thinning of the U-shaped base is the main driver for the shape transition, since it changes the non-dimensional torque. The behaviour is well modelled with a U-shaped plate with $d/h$ close to the critical value meaning a smooth transition with a large angle change, an optimal operating point for a transition that is reversible without hysteresis, as illustrated by a simulation of beam obeying this non-linear model in Figure 4(d).

## 5. Biomimetic programmable hinge

Taking inspiration from the nonlinear hinge of the *Spathiphyllum* plant, we designed a soft pneumatic actuator that mimics its remarkable bending properties. Here, we actively control the transverse curvature, whereas in the case of *Spathiphyllum*, it is rather the U-shaped cross section that varies in size with turgor pressure following a homothetic dilation. The structure is an elastomeric ribbon containing airways along its length that are off-centred (Figure 5(a)).

The structures were made of platinum-catalyzed silicone rubber (DragonSkin 10NV from Smooth-On) and were fabricated by mixing a prepolymer base and a curing catalyst in a 1:1 weight ratio. The liquid mixture was then poured on a 3D-printed mold (printed with a Prusa i3 MK3) and covered with a PMMA plate to ensure proper thickness of the sample. At the same time, a portion of the mixture was poured on a flat surface, yielding a 1.1-mm thick sheet. After curing (that typically takes 2 hours) and unmoulding, the structure was glued on the thick sheet using a very thin layer of the uncured mix of the same material as a glue. The structure was then pierced with a needle and connected to a pressure source, with a pressure sensor.

### 5.1. Pressurisation and transverse curvature

Upon pressurisation, the airways tend to change in cross-section, whereas their length remains unchanged (Siéfert et al., 2019). As the channels are off-centred, it generates a torque within the plate and hence transverse curvature in the ribbon in a similar way to common pneumatic soft robots (Shepherd et al., 2011) (Figure 5(a)).

The structure was inflated at various pressures and a picture was taken in the plane of the cross section. We computed the mean curvature $1/R$ of the ribbon by measuring the chord $\Delta$ and maximal deflection $\delta$. Basic geometry gives the relationship: $R = (\delta^2 + \Delta^2/4)/2\delta$.

To model the curvature resulting from the applied pressure in the airways, we adopt a bilayer approach (Timoshenko, 1925). We consider two layers in the structure, one of thickness $h - b$ containing centred airways of height $h_a$, and the other of thickness $b$. We then applied the model derived in Siéfert et al., (2019) to compute the target strain $\epsilon = f(\phi, \psi, p/E)$ within the top layer, where $\phi = w_a/(w_a + w_w)$ is the channel in-plane density, $\psi = h_a/(h - b)$ is the relative channel height, $p$ is the applied pressure and $E$ is the Young modulus of the material. Note that this model is nonlinear, as it computes the stresses in the deformed configuration. In a second step, an energetic bilayer approach is applied (Siéfert et al., 2021), considering a top layer with an inelastic strain $\epsilon$ and a bottom passive layer: the elastic energy is computed in each layer, assuming a cylindrical configuration with the curvature $1/R$ and the strain at the first layer midplane $\xi$, as the two unknowns:

$$U_1 = \frac{EwL}{2(1-\nu^2)} \left[ (1-\phi) \int_{-(h-b)/2}^{(h-b)/2} \left( \frac{z}{R} + \xi - \epsilon \right)^2 \mathrm{d}z \right.$$
$$\left. + 2\phi \int_{h_a/2}^{(h-b)/2} \left( \frac{z}{R} + \xi - \epsilon \right)^2 \mathrm{d}z \right] \tag{2}$$

$$U_2 = \frac{EwL}{2(1-\nu^2)} \int_{-h/2-b/2}^{-h/2+b/2} \left( \frac{z}{R} + \xi \right)^2 \mathrm{d}z. \tag{3}$$

The total energy $U_1 + U_2$ is then minimised with respect to $R$ and $\xi$ to get the curvature as a function of the strain $\epsilon$ in the top layer and the geometry. Again, the actual values of $h_a$, $H$ and $\phi$ in the deformed state are input in the formula to take into account the geometric nonlinearities.

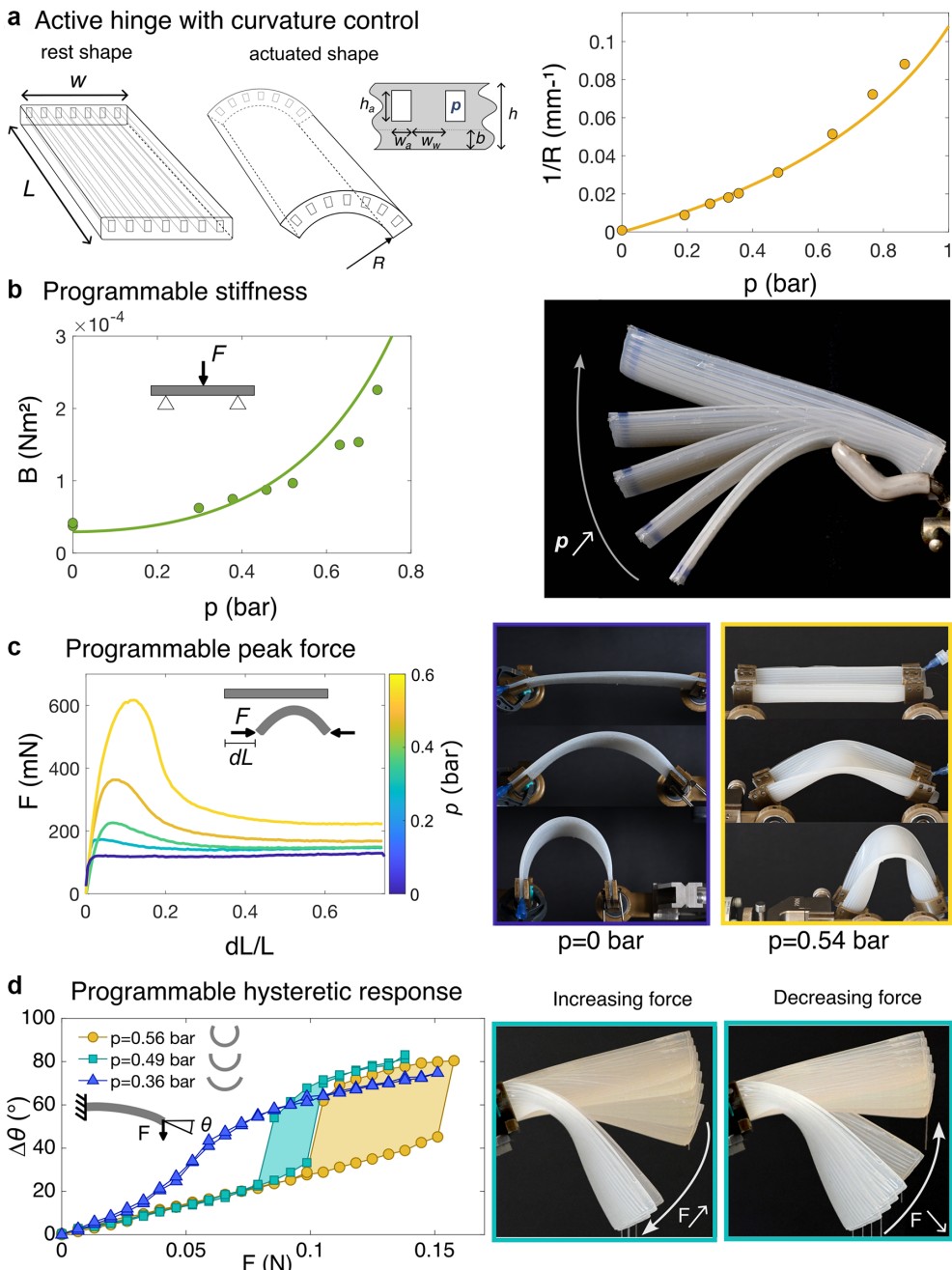

**Figure 5.** Active hinge inspired by the petiole base. (a) Left: the elastomeric structure contains off-centred airways along its length, that produces unidirectional curvature when pressurised. Right: curvature as a function of pressure for a structure with $L = 100$ mm, $w = 32$ mm, $h = 2.7 \pm 0.1$ mm, $b = 0.55 \pm 0.1$ mm, $h_a = 1.15 \pm 0.05$ mm, $w_a = 0.85 \pm 0.05$ mm, $w_w = 1.15 \pm 0.05$ mm, $E = 550 \pm 20$ kPa and $\nu = 0.5$. Circles correspond to experiments and the solid line to the model without any fitting parameter (Eqs. (2) and (3)).
(b) Programmable stiffness: when the pressure is increased, the transverse curvature induces a strong bending stiffening, such that the structure may sustain its own weight. Bending modulus, measured with a 3-point bending test, as a function of the inner pressure. Circles correspond to experiments and the solid line to the toy model (Eq. (B.1)), inferring the curvature computed in (a). (c) The nonlinear response may be also programmed. Upon compression, the structure buckles and exhibits a peak force that strongly varies with the transverse curvature and hence the inner pressure. Beyond this point, the force decreases to reach a plateau, that barely varies with the applied pressure.
(d) Programmable hysteretic response: cantilever experiment in which the end force is gradually increased and then decreased. As pressure is increased within the structure, it transitions from a smooth to a discontinuous behaviour with two jumps for the final orientation $\theta$, exhibiting a hysteretic loop that increases in size with the applied pressure. (right) Overlay of pictures with a regularly increasing (resp., decreasing) force, corresponding to the green curve, highlighting the sudden jump in orientation. For (c) and (d), structure with $w = 50$ mm, $h = 2.6 \pm 0.1$ mm, $b = 0.6 \pm 0.1$ mm, $h_a = 1 \pm 0.05$ mm, $w_a = 0.8 \pm 0.05$ mm, $w_w = 1.2 \pm 0.05$ mm, $E = 550 \pm 20$ kPa and $\nu = 0.5$. $L = 120$ mm for (c); for (d), $L = 105$ mm (yellow curve); $L = 80$ mm (green curve); $L = 60$ mm (blue curve).

## 5.2. Programmable bending stiffness

To measure the bending stiffness of the samples, we performed a 3-point bending test with a universal testing machine (Zwicki 0.5 kN from ZwickRoell). The two supports were separated by a distance of 57 mm. The force displacement curve was fitted linearly to get the stiffness and the bending stiffness was deducted using standard linear beam theory.

Controlling the transverse curvature enables us to increase by a factor of 6 the bending stiffness, as it strongly affects the apparent

thickness $d$ of the structure (see Eq. (B.1)). This dramatic increase in stiffness enables the structure to transition from drooping under its own weight to sustain it with minimal deflection as pressure is increased. Note that this effect is geometrical and results from the change in cross section and not from the pressure itself. The pressure within the airways is known to have a negligible impact on the bending stiffness of inflated elongated structures (Comer & Levy, 1963; Le Van & Wielgosz, 2005; Siéfert, 2019).

### 5.3. Compression of elastomeric samples

Beyond the linear stiffness, the nonlinear response may also be controlled by harnessing the opening of the U-shaped cross section upon loading.

The ribbon was compressed at various pressures using the universal testing machine in a horizontal configuration. 3D-printed clamps with a curvature matching approximately the transverse curvature of the inflated ribbon were mounted on ball bearings to ensure free rotation of the edges and a simply supported boundary condition. The compression test was performed at 200 mm/min.

When axially compressed, a flat simply supported ribbon first compresses while remaining flat but then buckles out of plane as the end-to-end distance is decreased (Figure 5(c)). This second step occurs at an almost constant load. For a transversely curved ribbon however, once the structure buckles out of plane, the force continues to increase as the end-to-end distance is decreased due to the high bending rigidity. The U-shaped cross section opens and the structures exhibit a peak force, beyond which the force decreases to reach a plateau (Figure 5(c)). Hence, the maximal force that the structure can sustain before dramatically collapsing may be actively adjusted by varying the internal pressure. Assembling such active elements in a metamaterial would lead to cellular structures with a tunable crushing load, realizing a versatile mechanical fuse.

### 5.4. Cantilever experiment

Additionally, the rich nonlinear behaviours described in Figure 4 may be reproduced with only one active structure: at low pressure, and hence low transverse curvature, the ribbon smoothly deflects as an end force is increased in a cantilever experiment (Figure 5(d), blue curve).

The ribbon was clamped at one edge using the curved 3D-printed clamps described above with an initial angle of 23° above horizontal. Paper clips of 0.67 g were then sequentially fixed on a small thread glued at the free end of the ribbon to increase the end force and a picture was taken from the side to measure the end orientation $\alpha$ of the ribbon. The clips were then removed in the unloading phase. The length of the cantilever was adjusted such that it barely sags under its own weight, when no additional end force was added. As the bending stiffness was highly dependent on the transverse curvature, the length was different in each experiment: $L = 105$ mm for $p = 0.56$ bar; $L = 80$ mm for $p = 0.49$ bar and $L = 60$ mm for $p = 0.36$ bar.

However, above a critical pressure, the deflection of the beam is discontinuous and a 'snap' occurs at a critical load. When unloading, another jump appears, but for a different load, revealing an hysteretic loop in the loading/unloading process. Both the critical snapping force and the size of the hysteretic loop may be tuned by adjusting the inner pressure in the structure (Figure 5(d), green and yellow curves). This bioinspired nonlinear hinge could be a useful building block for the design of tunable hysterons in computing mechanical metamaterials (Bense & van Hecke, 2021; Chen et al., 2021; Mei et al., 2021).

## 6. Conclusion

As a conclusion, we demonstrated that the change in shape of *Spathiphyllum* is due to a geometric mechanism, namely, the opening of U-shaped base under the weight load, softening the bending of the base and inducing a localised fold. This phenomenon is driven by the thinning of the petiole, rather than a decrease of the Young's modulus. This nonlinear mechanical system is likely to be on an optimal operating point for a reversible transition, featuring a high enough bending resistance to maintain the shape with minimum deflection over a large span of water content, while strongly and reversibly bending at a critical value of water loss. The same phenomenon was observed on other species featuring U-shaped petioles and water-rich tissues, suggesting that this sudden nonlinear drooping has evolved independently at least four times in angiosperms (Zuntini et al., 2024), specifically in Araceae, Asteraceae, Brassicaceae and Asparagaceae. This evolutionary convergence indicates the physiological relevance of this remarkable property: the sudden drooping on the floor at a critical water loss close to their physiological limit allows the leaf blades to hide from the sun and get closer to the ground (Chiariello et al., 1987; Gonzalez-Rodriguez et al., 2016; Zhang et al., 2010). This property relies on two collaborative effects: a U-shaped cross-section at the base of the petiole, that enables a nonlinear bending response, and a softening of the petiole, that may be driven by cross-section thinning or tissue softening, and triggers the drooping under almost constant weight load. Perspectives are to explore in more detail the precise role of the geometrical parameters of the U-shaped cross-section, especially the varying thickness, on the nonlinear bending response of the petiole. The precise effect of water loss on the shape and thinning of the cross-section is also yet to be rationalised.

## Appendix

### Appendix A: Model for the drying of cut leaves taking into account the shrinking section area

Water content in the petiole is followed by the relative weight fraction $\phi_w = M_w/(M_w + M_s)$, where $M_w$ and $M_s$ are the water and dry mass. The petiole piece after cutting is considered as cylindrical. The mass of water in a cylindrical piece is $M_w = \rho S L \phi_w$, with $\rho$ the density of the hydrated leaf (chosen to be that of water for numerical applications), $S = \pi d^2/4$ is the area of the cross-section of the petiole and $L$ is the length of the section. We assume the drying occurs by fluxes through the two cut surfaces of leaf pieces, with a rate proportional to the cut surface, inspired by Guo et al. (2024):

$$\frac{dM_w}{dt} = -2kS\phi_w \qquad \text{(A.1)}$$

with $k$ is a rate constant giving the flux of evaporation through the cut surface [water mass per unit time per unit area].

In the specific case of a constant cross-section, this equation rewrites into $d\phi_w/dt = -2k\phi_w/\rho L$, implying an exponential decay of the water weight fraction,

$$\phi_w = \phi_{w,0} \exp\left(-\frac{2k}{\rho L}t\right) \qquad \text{(A.2)}$$

and the evolution of the total mass is given by $M_{tot} = M_s/(1 - \phi_w)$ from the definition of $\phi_w$.

However, here the area of the cross section varied significantly over time, while the length of the cut sections did not evolve significantly (due to the presence of longitudinal fibres). We thus extend the previous analysis to include a variation of the cross-section area. The total volume of the cylindrical section is given by $V = M_{tot}/\rho = M_s/(1-\phi_w)/\rho$ and is now a function of $\phi_w$. The area of the section is thus $S = M_s/(1-\phi_w)/\rho L$, and the water mass is $M_w = [\phi_w/(1-\phi_w)]M_s$ from the definition of $\phi_w$. We therefore obtain from Eq. (A.1)

$$\frac{\mathrm{d}}{\mathrm{d}t}\left(\frac{\phi_w}{1-\phi_w}\right) = -\frac{2k}{\rho L}\frac{\phi_w}{1-\phi_w}.$$

It is solved by introducing the variable $u = \phi_w/(1-\phi_w)$, leading to $u = u_0 \exp(-2kt/\rho L)$, and therefore obtaining

$$\phi_w = \frac{u}{1+u} = \frac{\phi_{w,0}\exp(-2kt/\rho L)}{1-\phi_{w,0}+\phi_{w,0}\exp(-2kt/\rho L)}. \tag{A.3}$$

This yields a slightly more angular shape compared to an exponential decay (Eq. (A.2)), and agrees better with experiments (Figure 2(c) and Figure S3A in the Supplementary Material).

### Appendix B: Roman toy model, extension to non-isotropic tissues

The bending modulus of the U-shaped plate (assuming here for simplicity $\nu = 0$) is approximated for small transverse curvatures ($k_0 \ll 1/w$)

$$EI(k_0) = \frac{1}{12}Eh^3 w\left[1 + \beta\left(\frac{k_0}{\rho}\right)^2\right] \tag{B.1}$$

with $\beta = 1/60$. The first term in the brackets describes a flat plate ($k_0=0$), and the second one a curved thin plate. Indeed in the later case, in the cross-section plane with the $X$-axis on the flat plate and $Y$ the deviation, the neutral line is given by $Y_m = \int_{-w/2}^{w/2}(k_0 X^2)h\mathrm{d}X/hw = k_0 w^2/24$, so that the second moment of inertial is $I = \int_{-w/2}^{w/2}(k_0 X^2/2 - Y_m)^2 h\mathrm{d}X = k_0^2 hw^5/720$.

The bending energy writes $e_1 = EI(k_0)K^2/2$ per unit longitudinal length.

To account for the opening of the U-shape, the transverse curvature of the beam, initially $k_0$ can evolve to $k$, implying an energy cost $e_2 = Eh^3 w(k-k_0)^2/24$.

Adding the transverse and longitudinal bending energies (neglecting the crossed term, as done in Barois et al. (2014)), one obtains the total elastic energy:

$$e(K,k) = \frac{1}{24}Eh^3 w\left\{\left[1 + \beta\left(\frac{k}{\rho}\right)^2\right]K^2 + (k-k_0)^2\right\}. \tag{B.2}$$

We extend this model proposed by Ponomarenko (2012) and Roman (2024) to describe the case where the transverse bending costs less than the longitudinal bending, as observed in plants. This is achieved by adding a prefactor $\alpha$ in front of the transverse curvature energy $e_2$, resulting in:

$$e_m(K,k) = \frac{1}{24}Eh^3 w\left\{\left[1 + \beta\left(\frac{k}{\rho}\right)^2\right]K^2 + \alpha(k-k_0)^2\right\}, \tag{B.3}$$

with $\alpha = 1$ for an isotropic plate, while $\alpha < 1$ for an anisotropic plate, which is more easy to bend transversaly.

The transverse curvature adjusts to minimise the energy. We obtain equilibrium when $\partial e_m/\partial k = 0$ for $k = k_e$ such that

$$k_e = \frac{k_0}{1 + \frac{\beta}{\alpha}\frac{K^2}{\rho^2}}. \tag{B.4}$$

For this value of the transverse curvature, the energy writes

$$e_m = \frac{1}{2}\frac{Eh^3 w}{12}\rho^2 \frac{1 + \frac{\beta}{\alpha}(K/\rho)^2 + \beta(k_0/\rho)^2}{1 + \frac{\beta}{\alpha}(K/\rho)^2}(K/\rho)^2. \tag{B.5}$$

The couple exerted by a deformation is $M = \mathrm{d}e_m/\mathrm{d}K$ with

$$M = \frac{Eh^3 w}{12}K\frac{1 + 2\frac{\beta}{\alpha}(K/\rho)^2 + \beta(k_0/\rho)^2 + \frac{\beta^2}{\alpha^2}(K/\rho)^4}{\left[1 + \frac{\beta}{\alpha}(K/\rho)^2\right]^2}. \tag{B.6}$$

A crucial parameter is the initial transverse curvature $k_0/\rho$ normalised with $\rho = h/w^2$. For small transverse curvatures (small compared to $1/w$), simple geometry shows that this parameter tends to $k_0/\rho \simeq 4d/h$.

The shape of a beam under its own weight, with a weight per unit length $\mu$, can be simulated by solving the evolution of the torque along the curvilinear abscissa $s$: $\mathrm{d}M/\mathrm{d}s = -\mu g(s-L)\cos(\theta)$, with $\theta$ the angle of the tangent vector with respect to the horizontal, and L is the abscissa at the free end of the beam. Using the fact that the curvature is $K = \mathrm{d}\theta/\mathrm{d}s = \theta'$, and the fact that $\mathrm{d}M/\mathrm{d}s = \theta''\mathrm{d}M/\mathrm{d}K$ this equation is an ordinary differential equation on $\theta$ as a function of $s$. The boundary conditions for Figure 4(d) are a fixed angle of $\theta = 45°$ at the origin, and no torque at the end $K = \theta' = 0$.

### Appendix C: Wuest model

According to Wuest (1954), the torque needed to induce a longitudinal curvature $K$ of a U-shaped beam (same notations as the Roman toy model) is $M = \frac{Eh^3 w}{12(1-\nu^2)}\Big[K + \nu k_0 - \nu(k_0 + \nu K)F_1 + \frac{(k_0+\nu K)^2}{K}F_2\Big]$ with $F_1 = \frac{2}{\lambda}\frac{\cosh\lambda - \cos\lambda}{\sinh\lambda + \sin\lambda}$, $F_2 = \frac{1}{4}F_1 - \frac{\sinh\lambda\sin\lambda}{(\sinh\lambda+\sin\lambda)^2}$ and $\lambda = [3(1-\nu^2)]^{1/4}(K/\rho)^{1/2}$ (Seffen et al., 2000; Soykasap, 2007; Wuest, 1954). Note that in this model, the transverse curvature does not remain homogeneous across the width during the bending, contrary to the aforementioned toy model.

**Open peer review.** To view the open peer review materials for this article, please visit https://doi.org/10.1017/qpb.2025.10030.

### Acknowledgements

We warmly acknowledge Benoit Roman for sharing his toy model, and giving us the authorisation to use it to discuss our results. We also acknowledge fruitful discussion with Loïc Tadrist on the litterature of petioles.

**Competing interests.** There are no competing interests to declare.

**Author contributions.** C.Q., B.D., E.S. and P.M. designed research, B.B. developed the 1-N force machine, A.S., N.K., E.B. and E.S. performed research, A.S., N. K., E.S. and P.M. analysed data and P. M. and E.S. wrote the article. The first and second authors, A.S. and N.K., contributed equally to this work.

**Supplementary material.** The supplementary material for this article can be found at https://doi.org/10.1017/qpb.2025.10030.

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
