## [Reviewer Report]

This paper presents a combined experimental and theoretical study of the bending behavior of plant leaves when subject to water stress. The authors show convincing evidence that the drooping is caused by a ‘softening’ of the pulvinus, which behaves like a hinge. However, they also argue that this ‘softening’ is not caused by a change in the material properties as water content decreases, but instead that it is a geometric effect caused by the change in the cross-sectional area of the pulvinus. (Essentially, the moment of inertia, I, of the cross-section decreases significantly, so the bending stiffness B = E I decreases even if the modulus E remains the same.) The authors also discuss the nature of the drooping transition (whether it is hysteretic, or continuous) and suggest that remaining continuous would be advantageous for plants, to avoid damage.

The paper is generally well written, has a convincing story and would be of interest to readers of this journal. I just have a few minor comments that I think the authors should consider when revising the paper.

Scientific:

My main scientific comment concerns the description of the dichotomy between change of shape and softening - I think this is not very clearly expressed in the introduction. I think that authors such as Nilsson (and perhaps also the more recent paper by Chadler et al, which I noticed has now been officially published in R Soc. Interface) argued that it is the tension in the cell wall, which changes the effective bending stiffness – see also the work by Gao et al (doi: 10.1126/science.adi2997). Perhaps this could be rephrased to be clearer?

To make the main scientific point of the paper more clear, it would also be interesting to calculate the respective moments of inertia from the cross sections in figure 4A. Can the change in moment of inertia I, explain the large decrease in the bending stiffness without any change in E?

My other comments are more minor.

- I found the vertical colored lines in figure 2C very confusing. I think I eventually understood what the authors meant, but it would be clearer to me if the whole bottom half of the graph were shaded and labeled ‘reversible’ while the upper part were shaded a different color and labeled ‘irreversible’.

- Similarly, it wasn’t quite clear what angle they were referring to in figure 3C. Could this be shown somewhere?

- In figure 5D, I didn’t understand why the y-axis was F and x-axis was delta theta. Surely F is the control variable, so should be on the x-axis?

- I don’t think the negligible impact of pressure on the bending stiffness of elongated structures is that well known. However, the authors are right that it keeps coming up in the literature, usually when authors find that a pressure-dependent bending moment only occurs with wrinkling (see the references the authors mentioned, but also: doi:10.1103/PhysRevLett.128.058101 and 10.1016/0020-7683(94)90173-2). I think a clearer (but still brief) discussion of this point would be helpful.

- The analysis of Appendix B seems quite similar to Chapter 4 of the PhD thesis of Alexandre Ponomarenko (2012), which is entitled: “Écoulements critiques et plantes”, is publicly available via HAL, and was also undertaken in collaboration with Benoît Roman. It would be helpful if the authors could highlight any important differences for the reader, and perhaps cite this thesis too.

Linguistic:

- I did not understand the discussion about “Natural Intelligence” and using a “human brain to click on points” in section 2. This sounded like they were making an almost political point about not having used AI, but seems a bit unnecessary to me. A factual discussion of the experimental procedure would be clearer (and hence better).

- There are a few places where phrases that sound like English but I don’t think are, are used. For example ‘lineic’ should be ‘linear’, while I was not sure whether the use of ‘homothetically’ was intended to mean ‘hypothetically’ or ‘geometrically similar’. (I have not seen homothetically used before, but think it can mean geometrically similar – though the authors have not shown that this is the case here.)

---

## [Reviewer Report]

In the manuscript, the authors focused on the physical mechanisms on sudden but reversible drooping of the petiole under dehydration. They used Spathiphyllum whose petiole cross-section is U-shaped and contains a lot of water. They monitored plant’s shape and found that under sustained drought all petioles became more horizontal while the petiole remained quite straight. Experiments using transversal cross sections of the petiole at different locations indicated that the mass loss can reach more than 95% of the initial mass after dehydration. They also found that the mass loss range 0 to 0.3-0.4 where dehydration is reversible, while above irreversible tissue death starts to occur. During the mass loss, decay of width of the tissue was observed. Three-point bending measurements revealed that effective Young’s modulus did not seem to decrease on the physiologically reversible range 0 to 0.4 in water loss, the phase. From these observations, they hypothesized that thinning of the U-shaped part at the base of long petioles during drying are the key feature which explains the sudden drooping. They used models of the bending modulus of the U-shaped plate (Roman model and Wuest model). The model predicted non-linear response of the plate with torque. Using this idea, they built the bioinspired hinge.

The strong point of this manuscript is their use of multidisciplinary approaches (plant observation, modeling, and engineering biomimetic devices). Their approach may bring a new idea to plant scientists; why plants use U-shape petiole, or how they control their sudden movements in certain ambient environments, though it may be of interest to limited range of readers/study areas.

Here I would like to raise one major concern.

1. As the authors described “to explore in more detail the precise role of the geometrical parameters of the U-shaped cross-section, especially the varying thickness, on the nonlinear bending response of the petiole”, this point should be carefully assessed in the manuscript. As seen in Supplemental Figure 5, the U-shape looks “opened” during drying, not only reducing its thickness. Thus I suggest the authors need to check this effect before concluding as “the thinning of the U-shaped base is the main driver for the shape transition”.

Small another suggestion.

Section 2. “We used standard Natural Intelligence (NI) to monitor the position of the characteristic points of the petiole and leaf,”. Please provide more detailed information on this method.

---

## [Editor Report]

Both reviewers find this manuscript to be interesting and well-written. Some major and minor concerns to be addressed in the revision. Major revision is recommended.

---

## [Reviewer Report]

The authors have responded appropriately to my review. There are a couple of instances in the revised version where I found the language hard to parse:

At the bottom of page 1, what does the phrase “This was comforted on pneumatic cellular bioinspired materials Tadrist et al. (2022)” mean?

The authors have also introduced many more instances of the word ‘homothetic’ in the supplementary material, but without defining it. I think this would be useful in a journal such as this. (I still think ‘geometrically similar’ would be clearer.)

---

## [Reviewer Report]

The authors computed the second moment of inertia of the U-base from the images of a cross section. Based on their computation, it was concluded that size (cross-section area) is the main driving for this reduction, and not the evolution of the shape. The analysis was shown in Supplemental Figure 5. This additional experiment conducted by the authors was sufficient to convince the reviewer.

Minor issue

P3 line 4. “where I is the second moment of area,” This sentence is ended by “,”. Please modify this.